# Diverging likelihood of colon and rectal cancer in Yogyakarta, Indonesia: A cross sectional study

Herindita Puspitaningtyas[1], Susanna Hilda Hutajulu[2]*, Jajah Fachiroh[3], Nungki Anggorowati[4], Guardian Yoki Sanjaya[5], Lutfan Lazuardi[5], Patumrat Sripan[6]

**1** Faculty of Medicine, Public Health and Nursing, Doctorate Program of Health and Medical Science, Universitas Gadjah Mada, Yogyakarta, Indonesia, **2** Faculty of Medicine, Public Health and Nursing, Department of Internal Medicine, Division of Hematology and Medical Oncology, Universitas Gadjah Mada/ Dr. Sardjito General Hospital, Yogyakarta, Indonesia, **3** Faculty of Medicine, Public Health and Nursing, Department of Histology and Cell Biology, Universitas Gadjah Mada, Yogyakarta, Indonesia, **4** Faculty of Medicine, Public Health and Nursing, of Anatomical Pathology, Universitas Gadjah Mada, Yogyakarta, Indonesia, **5** Faculty of Medicine, Public Health and Nursing, of Health Policy and Management, Universitas Gadjah Mada, Yogyakarta, Indonesia, **6** Research Institute for Health Sciences, Chiang Mai University, Chiang Mai, Thailand

* susanna.hutajulu@ugm.ac.id

## Abstract

### Objectives

Colon and rectal cancer are associated with different risk factors and prognostic. However, this discrepancy has not been widely explored in the local population. This study aimed to investigate the site-specific likelihood of colorectal cancer (CRC) incidence in the Yogyakarta province, Indonesia.

### Methods

This cross-sectional study analyses 1,295 CRC cases diagnosed in 2008–2019 registered in the Yogyakarta population-based cancer registry (PBCR) database. Cases were grouped into colon and rectal cancer. Log-binomial regression was used to determine the relative risk of either colon or rectal cancer across different gender, age group, and rurality of residence. The age-specific rates were calculated by age group and temporal trend for each group were analyzed using joinpoint regression.

### Results

Females displayed higher odds of colon cancer (relative risk/RR = 1.20, 95%CI = 1.02–1.41) and lower odds of rectal cancer (RR = 0.92, 95%CI = 0.85–0.99). Elevated odds of colon cancer were observed in younger age group, especially 30–39 (RR = 1.87, 95%CI = 1.10–3.19), while decreased odds of rectal cancer was apparent in age group 30–39 and 40–49 (RR = 0.75, 95%CI = 0.60–0.93 and RR = 0.82, 95%CI = 0.69–0.98, respectively). Living in urban or rural areas did not significantly influence the odds of either having colon (RR = 0.98, 95%CI = 0.82–1.17) or rectal cancer (RR = 1.01, 95%CI = 0.93–1.10). During

**Data Availability Statement:** Data cannot be shared publicly because of restrictions imposed by the ethics committee as most of these contain patient data, albeit de-identified, and it may be

possible to determine the identity of participants. Should there be a request for data, this can be sent to the corresponding author (email: susanna. hutajulu@ugm.ac.id) orthe institutional ethics committee (email: mhrec_fmugm@ugm.ac.id) at Universitas Gadjah Mada, Indonesia, with data access queries as well.

**Funding:** SHH received funding from the The Indonesian Ministry of Research, Technology, and Higher Education (2023) under grant number 2140/UNI/DITLIT/Dit.Lit/PT.01.03/2023. Funder has no role in the study design, data collection and analysis, decision to publish, or preparation of the manuscript.

**Competing interests:** The authors have declared that no competing interests exixst.

2008–2019, trends of colon cancer in age <50 increased by 8.15% annually while rectal cancer displayed a 9.71% increase annually prior to 2017, followed by a 17.23% decrease until 2019.

## Conclusions

Yogyakarta population shows higher odds of young-onset colon cancer, especially between age 30–39 years old. Overall observation of trend shows increasing incidence in young-onset colon cancer, and non-significant decrease in rectal cancer.

## Introduction

Colorectal cancer (CRC) is the third most prevalent malignancy and the second leading cause of cancer death worldwide [1]. The Asian population ranks first in terms of the number of incidences and mortality of CRC compared to other populations worldwide [2]. The trend of CRC is expected to increase across different populations in the world, both in the West, including the United States, Canada, and Australia, and in Eastern populations, such as China, and Korea [3].

In Indonesia, 34,189 new cases of CRC were estimated in 2020, ranking the fourth most prevalent cancer incidence in the country [4]. The CRC mortality rate is the fifth largest, reaching 6.7 deaths per 100,000 person-years [1]. In Yogyakarta Province, one of 34 provinces in Indonesia with the highest cancer prevalence according to the National Basic Health Survey [5], CRC incidence was recorded to be 7.40 per 100.000 population [6]. Colorectal cancer cases in Indonesia were dominated by males (54%), with a peak incidence at 50–54 years of age [7]. The local cases reported were also dominated by advanced tumors (30–40%) [6,8], all of which were associated with worse prognosis and decline in quality of life.

Colon and rectal cancer are frequently regarded collectively as colorectal cancer due to the assumption that they arise from organs that have similar morphological and molecular characteristics. However, the colon and rectum possess various differences in terms of their histological and anatomical origin, physiological properties, and genetic pathways, attributing to the different risks and survival among the two cancers [9,10]. The current estimate in the Global Cancer Observatory shows a higher incidence of colon compared to rectal cancer (11.4 vs. 7.6 per 100,000 population worldwide) [1].

Histological differences might explain the lower incidence of rectal cancer, including the better defense provided by the mucin-secreting goblet cells that present in higher quantities in the rectum, resulting in the lower cases reported on the respective site [10]. In addition, colon cancer has a flat morphology, making it more frequently diagnosed at a later stage and with poor differentiation [11]. Alteration in mismatch repair (MMR) genes in the germinal cells might result in hereditary nonpolyposis CRC (HNPCC), which is predominant in the distal colon and rectum, and involved familial adenomatous polyposis (FAP). When this occurs in somatic cells, this would result in microsatellite instability (MSI) that is responsible for the oncogenesis of sporadic CRC and is predominant in the proximal colon [10,11]. The different characteristics result in different response to treatment and prognosis. Patients with lower MSI, usually in cases of rectal cancer, have a better prognosis with adjuvant chemotherapy and targeted therapy [11]. Another study observed a worse prognosis of rectal cancer in the presence of complications compared to colon cancer [12].

The difference in risks between colon and rectal cancer is also influenced by other factors such as age, sex, and location of residence [13–15]. Distal colon and rectal cancer, with a higher proportion of chromosomal instability (CIN)-high tumors, are usually predominated by male sex and younger age. On the contrary, proximal colon cancer which is more frequently present with MSI-high tumors, are predominant in female and older age [11].

The disparity of risk among different sexes might be attributable to sexual dimorphism, due to the numerous mechanisms of protective effects of estrogens against colon and rectal cancer in females [14]. However, previous studies have suggested the existence of gender-specific risk of colon and rectal cancer due to the difference in anatomical characteristics, resulting in lower sensitivity of screening tools among females, hence worse presentation and prognosis [16]. According to the more recent consensus molecular subtypes (CMSs) stratification by the CRC Subtyping Consortium in 2015, females with right-sided colon cancer were more frequently diagnosed with CMS1 and were also associated with higher histological grade and worse prognosis after relapse. Conversely, male, who were more frequently diagnosed with left-sided MSI2 tumors, has notably better prognosis [13].

The risk of rectal cancer is suggested to be 1.5 higher in males, while the risk of colon cancer is 1.2 in females [9]. However, the incidence of CRC consistently demonstrated a higher burden among males compared to females, both in colon and rectal cancer [3,15]. At present, the worldwide incidence of colon cancer (10 per 100,000) almost doubles those of rectal cancer (5.6 per 100,000) among women (H. Sung et al., 2021). Moreover, the disparity among sexes in rectal cancer is more than twice that observed in colon cancer (male-to-female ratio = 1.75 vs. 1.31, respectively) [17].

Age also contributes to other major non-modifiable risk factors for both colon and rectal cancer, with a significant increase in incidence and mortality after the age of 50. Around 60% of global CRC incidence and 50% of its mortality presented between the ages of 50 to 74 years [18,19]. Although the reason is yet fully understood, the proportion of early-onset CRC (EOCRC) is increasing and recorded higher, especially among Asian countries [19]. EOCRC is more frequently present at the rectal location, with a more advanced stage, and more unfavorable histopathology, worsening its prognosis [20,21]. In younger age, MSI occurrence in rectal cancer strongly indicates a mutation associated with HNPCC or Lynch syndrome that usually shows MSI-high status [11,22].

Several other modifiable lifestyle factors such as high BMI and physical inactivity are frequently associated with an increased risk of colon cancer, but not rectal cancer, emphasizing the difference in etiology between the two tumor sites [11]. Living in urban areas has been found to be associated with a higher risk of different types of cancer [23]. CRC, however, were among the cancers with a higher incidence in rural areas, owing to the lower coverage of screening in rural areas [23–25]. Another study suggests that the association of rurality on the incidence of CRC might also be associated with other factors such as sex, race, and tumor location [23].

At present, the difference in risk between colon and rectal cancer and its association with differences between sex, age, and rurality of residence has been studied in several different populations [16,26–28]. This, however, has not been widely studied in Indonesia. In the present study, we aim to examine the influence of sex, age, and different risks attributable to location of living to the risk of colon and rectal cancer and to describe the trend of CRC in the local population in Yogyakarta Province.

## Materials and methods

### Study design and sample

We performed a retrospective study using secondary data from the Yogyakarta Population-based Cancer Registry (PBCR) to examine different risks of colon and rectal cancer among the

population of the Yogyakarta Province, Indonesia. Cases in the Yogyakarta PBCR come from three out of the five districts in Yogyakarta Province, namely Sleman, Yogyakarta City, and Bantul District. Collectively, the three districts comprised 48 sub-districts with a total of 2.8 million population in 2023 [29]. Previous study reported CRC incidence in Yogyakarta Province to be 7.40, ranging 9.20 in Yogyakarta City, 7.07 in Bantul, and 7.03 per 100,000 population in Sleman [6].

We extracted all primary CRC cases (ICD-10 C18.0–9, 19.9, and 20.9 from the Yogyakarta PBCR database, diagnosed from 2008–2019 (n = 1,597). All cases recorded were residents of the three catchment areas, who have lived in the region for at least consecutive six months prior to diagnosis. Data extraction was done in June 2023 and all data retrieved were de-identified. We obtained sociodemographic and clinical data on the cases, including sex, age, location of residence, histopathological type, and stage of the tumor. The right colon category includes caecum, ascending colon, hepatic flexure, and transverse colon. The left colon category includes the descending colon, splenic flexure, and sigmoid colon. For the purpose of analysis, cases with unspecified location (C18.8–9) were excluded (n = 302, 18.9%). A total of 1,295 cases were included in the final analysis, with overall percentage of morphological verification (% MV) of 77.0% (S1 Table).

## Data analysis

To observe the difference in likelihood in the population, CRC cases were classified in colon cancer group, including cases of right colon (C18.0–4) and left colon (C18.5–7), and rectal cancer group, including rectosigmoid junction (C19.9) and rectal (C20.9) topography. Clinical characteristics of the cases such as sex, age, location of residence, and stage of cancer were retrieved from the registry database. Crude and adjusted relative risk of colon and rectal cancer were calculated using the log-binomial regression model. Age was stratified in 10-year increments and the ≥80 years old age group was used as the reference category. The status of the rurality of the residence was determined based on the subdistrict where the cases were residing, according to the classification of the national government [30]. Crude relative risk (RR) with a 95% confidence interval (95%CI) was calculated using the bivariate log-binomial regression model. Multivariable log-binomial regression analysis was employed to determine the adjusted relative risk (ARR) of either having colon or rectal cancer. P-values were estimated by two-sided tests and significance was set at p-value <0.05.

For the temporal analysis, we sourced population data for Sleman, Kota Yogyakarta, and Bantul districts in the year 2014 from the Indonesian Central Bureau of Statistics as the denominator (S2 Table). It was used to determine the age-standardized incidence rates (ASRs) of 2008–2019 as the central year in the study timeframe. The World Standard Population by Segi was used as a weight reference in computing the ASR [31]. ASRs are reported per 100,000 person-years and are stratified into three groups: <50 or early-onset colorectal cancer (EOCRC), 50–64, and ≥65 years old. ASR was computed in each year observed and each age group for all cases of CRC combined as well as colon cancer and rectal cancer subgroups. To further analyze the temporal trends in incidence rates among the age groups, we employed the joinpoint regression test to compare the annual trends within each age group and across the specified subgroups.

All statistical analysis was conducted using STATA version 17.0 (StataCorp LLC., Texas, TX, USA). The temporal trend was observed using Joinpoint Regression Program version 5.0.2 (SEER, USA). The present study used secondary data from the Yogyakarta PBCR that obtained data from multiple health facilities, including primary health care and referral hospitals under a national decree from The Indonesian Ministry of Health and further provincial

clearance. In each health facility, each patient has given written general consent upon admission, including consent for data utilization for research purposes. The use of the de-identified secondary data in this study was acknowledged and approved by the Joint Ethical Committee of the Faculty of Medicine, Public Health, and Nursing, Universitas Gadjah Mada and Dr. Sardjito General Hospital (KE/FK/0771/EC/2023).

## Results

### Characteristics of colon and rectal cancer patients

The present study performed an analysis of the demographic and clinical characteristics of a cohort of 1,295 CRC patients drawn from the Yogyakarta PBCR database, as detailed in Table 1. More cases in the cohort were present with rectal cancer (n = 893, 68.96%). A male predominance was present in the rectal cancer subgroup (n = 504, 56.44%). In the case of

**Table 1. Characteristics of CRC patients in Yogyakarta population-based cancer registry (n = 1,295).**

| Variables | CRC N (%) | Site of tumour | | p-value |
| --- | --- | --- | --- | --- |
| | | Colon cancer N (%) | Rectal cancer N (%) | |
| Number of cases | 1,295 (100) | 402 (31.04) | 893 (68.96) | |
| Sex | | | | 0.025* |
| Male | 704 (54.36) | 200 (49.75) | 504 (56.44) | |
| Female | 591 (45.64) | 202 (50.25) | 389 (43.56) | |
| Age | | | | |
| Mean, years (SD) | 57.56 (13.61) | 55.99 (13.43) | 58.27 (13.63) | 0.058 |
| <20 years | 3 (0.23) | 1 (0.25) | 2 (0.22) | |
| 20–29 years | 36 (2.78) | 9 (2.24) | 27 (3.02) | |
| 30–39 years | 100 (7.72) | 42 (10.45) | 58 (6.49) | |
| 40–49 years | 190 (14.67) | 69 (17.16) | 121 (13.55) | |
| 50–59 years | 372 (28.73) | 119 (29.60) | 253 (28.33) | |
| 60–69 years | 339 (26.18) | 96 (23.88) | 243 (27.21) | |
| 70–79 | 197 (15.21) | 53 (13.18) | 144 (16.13) | |
| 80+ | 58 (4.48) | 13 (3.23) | 45 (5.04) | |
| Residence | | | | 0.830 |
| Rural | 407 (31.43) | 128 (31.84) | 279 (31.24) | |
| Urban | 887 (68.57) | 274 (68.16) | 614 (68.76) | |
| Tumor location | | | | 0.000 |
| Right colon | 170 (13.13) | 170 (42.29) | | |
| Left colon | 232 (17.92) | 232 (57.71) | | |
| Rectosigmoid | 205 (15.83) | | 205 (22.96) | |
| Rectum | 688 (53.13) | | 688 (77.04) | |
| Stage | | | | 0.003** |
| Stage 1 | 39 (3.01) | 13 (3.23) | 26 (2.91) | |
| Stage 2 | 65 (5.02) | 31 (7.71) | 34 (3.81) | |
| Stage 3 | 83 (6.41) | 35 (8.71) | 48 (5.38) | |
| Stage 4 | 164 (12.66) | 52 (12.94) | 112 (12.54) | |
| Unknown | 944 (72.90) | 271 (67.41) | 673 (75.36) | |

Abbreviation: CRC = colorectal cancer, SD = standard deviation.

*p<0.05

**p<0.005.

colon cancer, a near-equivalent sex distribution was observed, with 49.75% males (n = 200) and 50.25% females (n = 202). The mean age at presentation for the entire cohort was 57.66 years, with a slightly younger mean age noted in colon cancer patients (mean = 55.99±13.43) compared to those with rectal cancer (mean = 58.27±13.63, p = 0.06). The age of subjects ranged from 14 to 93 with most subjects diagnosed in the 50–59 age group, both in colon (n = 119, 29.60%) and rectal (n = 253, 28.33%) cancer. The proportion of young-onset cases (<50 years) or EOCRC were 25.41%, comprises of 121 cases in the colon (30.10%) and 208 cases (23.29%) in the rectal cancer group. In both groups, more than 68.57% of subjects resided in urban areas within the three districts.

## Comparison of colon and rectal cancer by sex and age

We presented the result of the log-binomial regression analysis for colon and rectal cancer in Table 2. In this study, female patients exhibit 20% higher odds of colon cancer as compared to males (RR = 1.20, 95%CI = 1.02–1.44, p = 0.025). Moreover, within the stratified age group, subjects aged 30–39 years old have an 87% higher likelihood of colon cancer (RR = 1.87, 95% CI = 1.10–3.19, p = 0.021). Our finding observed no discernible disparity in the odds of developing colon cancer among those residing in urban compared to rural areas in this population (RR = 0.98, 95%CI = 0.82–1.17, p = 0.830).

Our observation indicates an 8% lower likelihood of rectal cancer among female subjects as compared to the males (RR = 0.92, 95%CI = 0.85–0.99, p = 0.027). In contrast to our finding on the colon cancer subgroup, subjects aged 30–39 and 40–49, respectively, demonstrate a 25% and 18% decrease in the odds of rectal cancer (RR = 0.75, 95%CI = 0.60–0.93, p = 0.008 and RR = 0.82, 95%CI = 0.69–0.98, p = 0.027). Similar to their counterparts in the colon cancer subgroup, residing in urban areas also did not significantly elevate the odds of rectal cancer (RR = 1.01, 95%CI = 0.93–1.10, p = 0.831).

**Table 2. Crude and adjusted relative risk of colon and rectal cancer among CRC cases in Yogyakarta.**

| Predictors | Colon | | | | | | | | Rectal | | | | | | | |
|---|---|---|---|---|---|---|---|---|---|---|---|---|---|---|---|---|
| | RR | 95%CI | | p-value | ARR | 95%CI | | p-value | RR | 95%CI | | p-value | ARR | 95%CI | | p-value |
| | | lower | upper | | | lower | upper | | | lower | upper | | | lower | upper | |
| Gender | | | | | | | | | | | | | | | | |
| Male | ref. | | | | ref. | | | | ref. | | | | ref. | | | |
| Female | 1.20 | 1.02 | 1.41 | 0.025* | 1.21 | 1.03 | 1.42 | 0.022* | 0.92 | 0.85 | 0.99 | 0.027* | 0.93 | 0.86 | 1.00 | 0.043* |
| Age | | | | | | | | | | | | | | | | |
| <20 | 1.48 | 0.28 | 7.90 | 0.641 | 1.47 | 0.27 | 8.06 | 0.859 | 0.71 | 0.38 | 1.94 | 0.714 | 0.83 | 0.36 | 1.91 | 0.668 |
| 20–29 | 1.12 | 0.53 | 2.34 | 0.773 | 1.12 | 0.53 | 2.34 | 0.766 | 0.97 | 0.77 | 1.22 | 0.776 | 0.96 | 0.76 | 1.21 | 0.731 |
| 30–39 | 1.87 | 1.10 | 3.19 | 0.021* | 1.88 | 1.11 | 3.18 | 0.020* | 0.75 | 0.60 | 0.93 | 0.008* | 0.75 | 0.61 | 0.93 | 0.009* |
| 40–49 | 1.63 | 0.97 | 2.71 | 0.066 | 1.62 | 0.97 | 2.69 | 0.066 | 0.82 | 0.69 | 0.98 | 0.027* | 0.82 | 0.69 | 0.97 | 0.022* |
| 50–59 | 1.42 | 0.86 | 2.36 | 0.164 | 1.42 | 0.86 | 2.35 | 0.165 | 0.88 | 0.75 | 1.02 | 0.096 | 0.87 | 0.75 | 1.02 | 0.082 |
| 60–69 | 1.26 | 0.76 | 2.10 | 0.367 | 1.25 | 0.76 | 2.08 | 0.383 | 0.92 | 0.79 | 1.08 | 0.313 | 0.92 | 0.79 | 1.07 | 0.277 |
| 70–79 | 1.20 | 0.71 | 2.04 | 0.501 | 1.20 | 0.71 | 2.03 | 0.505 | 0.94 | 0.80 | 1.11 | 0.471 | 0.93 | 0.79 | 1.09 | 0.360 |
| 80+ | ref. | | | | ref. | | | | ref. | | | | ref. | | | |
| Residence | | | | | | | | | | | | | | | | |
| Rural | ref. | | | | ref. | | | | ref. | | | | ref. | | | |
| Urban | 0.98 | 0.82 | 1.17 | 0.830 | 0.98 | 0.83 | 1.17 | 0.831 | 1.01 | 0.93 | 1.10 | 0.831 | 1.01 | 0.94 | 1.10 | 0.725 |

Abbreviation: CRC = colorectal cancer, RR = risk ratio, ARR = adjusted risk ratio.

*p<0.05.

In the multivariable analysis, sex remains significantly associated with the likelihood of colon cancer, in which female subjects demonstrated a 21% increase in odds, controlling for age and location of residence (ARR = 1.21, 95%CI = 1.03–1.42, p = 0.022). Subjects in the 30–39 age group have an 88% significant increase in the probability of developing colon cancer compared to those aged 80 and older (ARR = 1.88, 95%CI = 1.11–3.18, p = 0.020). Despite not being statistically significant, we observed higher odds of developing colon cancer in all age groups when compared to the oldest group, controlling for sex and location of residence.

A contrasting, and consistent, result was evident in the multivariable analysis result of the rectal cancer subgroup. Female subjects displayed 7% lower odds compared to males, controlling for age and residence (ARR = 0.93, 95%CI = 0.86–1.00, p = 0.043). Subjects in the 30–39 and 40–49 age groups displayed similarly reduced odds of developing rectal cancer when controlling for age and residence (RR = 0.75, 95%CI = 0.61–0.93, p = 0.009 and RR = 0.82, 95% CI = 0.69–0.97, p = 0.022).

## The age-stratified trend of colon and rectal cancer

The analysis of dynamics on the annual change in the trend of the overall CRC as well as colon and rectal cancer is presented in Fig 1 for cases younger than 50 years (left), age group 50–64 (middle), and age group 65 and older (right). In subjects younger than 50, we observed a significant increase in the trend of CRC from 2008 to 2017 (annual percent change/APC = 7.38%, 95%CI = 2.93–13.03%). Similarly in colon cancer, the trend of incidence increases 8.15% annually on average (95%CI = 1.62–16.94%). In rectal cancer, however, the increasing trend from 2008–2017 (APC = 9.71%, 95%CI = 6.49–27.33%) was followed by a non-significant decrease from 2017–2019 (APC = -17.23%, 95%CI = -17.23–32.50%).

A notable significant increase in CRC incidence was apparent in cases aged 50–64 (Fig 1 (middle), APC = 16.93%, 95%CI = 13.39–22.84%), from 4.79 per 100,000 population in 2008 to 23.61 per 100,000 population in 2019. In the same age group, a similar trend was observed for colon (APC = 15.30%, 95%CI = 7.75–29.11%) and rectal cancer (APC = 17.08%, 95% CI = 13.02–23.32%). Despite not being as high as the 50–64 age group, we also observe a similar increase in the trend of CRC in the 65 and older group (APC = 11.38%, 95%CI = 7.64–16.76%), from 9.22 per 100,000 population in 2008 to 24.73 per 100,000 population in 2019. In this age group, the increase in trend was very similar between those in the colon (APC = 10.90%, 95%CI = 1.63–24.82%) and rectal cancer (APC = 10.95%, 95%CI = 4.09–21.26%).

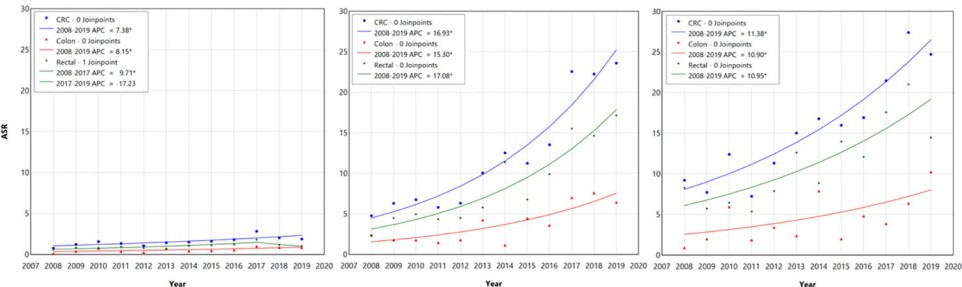

**Fig 1.** Average annual change of colorectal cancer (blue), colon cancer (red), and rectal cancer (green) incidence among subjects aged under 50 years old (left), 50–64 years old (middle), and 65 years old or older (right). CRC incidence demonstrated an upward trend across the three different age groups. The increase of rectal cancer is higher among the under 50 and 50–64 age group, compared to colon cancer. The increase of colon and rectal cancer however is similar in the 65 and older age group.

## Discussion

In the present study, we observed a higher proportion of rectal cancer (68.96%) in the studied population. This finding is contradictory to the global data showing the higher incidence cases of colon compared to rectal cancer [1]. Although this might be attributable to the high proportion of cases with unspecified location (ICD-10 C18.8–9) excluded from the study (18.9%), our result is similar to the proportion of colon and rectal cancer in previous studies coming from low- and middle-income countries [32,33]. This proportion is also similar to Vietnam (59.3%) and North Korea (51.7%) which reported a higher number of incidence rectal cases compared to colon [1].

Our data shows higher odds of colon cancer and a lower likelihood of rectal cancer among females. This is also supported by previous data suggesting a 1.2-fold risk of colon cancer and 0.5 risk of rectal cancer among females [9]. Another study also highlights the elevated risk of colon cancer among the female sex, especially in the proximal colon [34]. This difference could be explained by the presence of higher insulin susceptibility in the colon. Changes in the expression of insulin receptors trigger abnormal cell proliferation that may induce malignancy [35]. The high concentration of estrogen in females further promotes this mechanism, resulting in a higher risk of colon cancer in women [34,36].

We observed 30% of patients with colon cancer and 23% with rectal cancer diagnosed before the age of 50. This finding is consistent with a previous observation performed in the Tanzania population [32]. This is also in line with our finding, showing an increase in the overall trend of CRC and colon cancer, and a decline in the trend of rectal cancer in the last three years among the younger group. This overall proportion of EOCRC cases in this study (25.4%) is comparably higher than previously reported in the Western population which ranged less than 10% as reported from Milan, Italia and Florida, USA [19,27]. This proportion was also higher than reported from neighboring Southeast Asian countries, ie. 14.5% in Northern Malaysia and 5.7–13.4% in Singapore, as well as the 30–40% proportion previously reported from pathology and hospital-based data from Indonesia, demonstrating the higher incidence of EOCRC in the Asian population [8,37–40].

The increasing risk of CRC among the younger population has been observed consistently among different populations [27,41–44]. The higher frequency of Lynch syndrome in the local population possibly contributes to the higher EOCRC compared to the Western population. The proportion of cases with probable Lynch syndrome as the underlying cause in the local population is reported to be 6-fold of those reported in western countries [45]. Another previous study from Indonesia also reported the proportion of Lynch syndrome among EOCRC almost two-fold of those with older age of onset [42].

The risk of developing EOCRC was associated with lifestyle factors such as low physical activity, sedentary behavior, smoking and high processed meat consumption [46–49]. Despite the changing profile of gut-microbiota along with ageing, higher presentation of microbiota is associated with both the initiation, promotion, and progression of CRC oncogenesis among younger cases [50]. Although not observed in the present study, the association of age on the risk of colon and rectal cancer is stronger among women, who have higher odds of presenting with tumor characteristics associated with worse prognostic [34].

Although not further observed in the present study, caution needs to be taken when interpreting the gap of incidence between EOCRC cases and the older age groups. It has been observed that EOCRC might have longer symptom duration, time to diagnosis and treatment as compared to older cases, especially in cases with less advanced stages [51]. The fact that CRC screening is currently only recommended to those with known familial syndrome and familial history [52], might undermine the actual burden among younger age due to low diagnostic coverage.

The present study is the first to use population-based data to evaluate the risk of colon and rectal cancer in the local population, which can provide stronger evidence of the actual disease burden as compared to hospital-based data as previous reports from Indonesia. This study provides evidence of the high and increasing burden of young-onset CRC in the local population, suggesting the need to also focus on the younger population in an effort to the control cancer burden. The inclusion of only data from the Yogyakarta Cancer Registry in this study warrants further exploration utilizing data from a wider scope of population to capture the condition of other provinces in Indonesia that possess variations in lifestyle, diet, as well as environmental risk factors.

## Conclusions

Our findings highlight the difference in relative risk of colon and rectal cancer, especially in the younger age group. Some of these patterns need more detailed exploration to understand other risk factors that might influence this trend. The fact of higher risk of young onset colon cancer, especially in between age 30–49 years old, may serve as recommendation increase effort and awareness for screening especially towards young population. A nationwide investigation may provide better and more comprehensive data for calling for actions in various levels of authority in performing CRC screening programs and further promotion of healthcare services to provide better management for patients with CRC.

## Supporting information

**S1 File. Minimal dataset.**
(XLSX)

**S1 Table. Parameter of data quality of colorectal cancer cases in the Yogyakarta PBCR year 2008–2019.**
(PDF)

**S2 Table. Population denominator of the Yogyakarta PBCR.**
(PDF)

## Acknowledgments

The authors thank Yogyakarta population-based cancer registry and Dr. Sardjito hospital-based cancer registry for providing data used in the present analysis.

## Author Contributions

**Conceptualization:** Herindita Puspitaningtyas, Susanna Hilda Hutajulu, Jajah Fachiroh.

**Data curation:** Herindita Puspitaningtyas.

**Formal analysis:** Herindita Puspitaningtyas, Susanna Hilda Hutajulu, Jajah Fachiroh, Patumrat Sripan.

**Funding acquisition:** Susanna Hilda Hutajulu.

**Investigation:** Herindita Puspitaningtyas, Susanna Hilda Hutajulu, Nungki Anggorowati.

**Methodology:** Herindita Puspitaningtyas, Susanna Hilda Hutajulu, Jajah Fachiroh, Patumrat Sripan.

**Project administration:** Herindita Puspitaningtyas, Susanna Hilda Hutajulu.

**Resources:** Herindita Puspitaningtyas, Susanna Hilda Hutajulu, Nungki Anggorowati.

**Software:** Herindita Puspitaningtyas, Guardian Yoki Sanjaya, Lutfan Lazuardi.

**Supervision:** Susanna Hilda Hutajulu, Jajah Fachiroh, Nungki Anggorowati, Guardian Yoki Sanjaya, Lutfan Lazuardi.

**Validation:** Herindita Puspitaningtyas, Susanna Hilda Hutajulu, Nungki Anggorowati.

**Visualization:** Herindita Puspitaningtyas, Susanna Hilda Hutajulu.

**Writing – original draft:** Herindita Puspitaningtyas, Susanna Hilda Hutajulu.

**Writing – review & editing:** Herindita Puspitaningtyas, Susanna Hilda Hutajulu, Jajah Fachiroh, Nungki Anggorowati, Guardian Yoki Sanjaya, Lutfan Lazuardi, Patumrat Sripan.

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
