## [Decision Letter · Decision Letter 0]

14 Feb 2024

PONE-D-23-37906Diverging likelihood of colon and rectal cancer in Yogyakarta, Indonesia: a cross sectional studyPLOS ONE

Dear Dr. Hutajulu,

Thank you for submitting your manuscript to PLOS ONE. After careful consideration, we feel that it has merit but does not fully meet PLOS ONE’s publication criteria as it currently stands. Therefore, we invite you to submit a revised version of the manuscript that addresses the points raised during the review process.

We look forward to receiving your revised manuscript.

Kind regards,

Nontuthuzelo Iris Muriel Somdyala, Ph.D

Academic Editor

PLOS ONE

Journal Requirements:

3. In the online submission form, you indicated that Data cannot be shared publicly because of restrictions imposed by the ethics committee as most of these contain patient data, albeit de-identified, and it may be possible to determine the identity of participants. Should there be a request for data, this can be sent to the corresponding author (email: susanna.hutajulu@ugm.ac.id)

4. You have indicated that data is available from [email: mhrec_fmugm@ugm.ac.id].  Please can we ask you to provide us with a general contact email address for the data requests, so readers can request access in perpetuity. If a general email is not available please provide a link to a website where readers can obtain access to data. 

5. Please respond by return e-mail with an updated version of your manuscript to amend either the abstract on the online submission form or the abstract in the manuscript so that they are identical. We can make any changes on your behalf.

Reviewers' comments:

Reviewer's Responses to Questions

**Comments to the Author**

1. Is the manuscript technically sound, and do the data support the conclusions?

Reviewer #1: Yes

2. Has the statistical analysis been performed appropriately and rigorously? 

Reviewer #1: Yes

3. Have the authors made all data underlying the findings in their manuscript fully available?

Reviewer #1: No

4. Is the manuscript presented in an intelligible fashion and written in standard English?

Reviewer #1: Yes

5. Review Comments to the Author

Reviewer #1: - Please provide results of the indices of data quality of the PBCR

- Please provide a reference for the denominator.

- Please mention the standard population with a reference used for the calculation of standardized rates

6. PLOS authors have the option to publish the peer review history of their article (what does this mean?). If published, this will include your full peer review and any attached files.

Reviewer #1: No

---

## [Author Response · Author response to Decision Letter 0]

27 Feb 2024

Dear Editor and Reviewer,

We are thankful for the positive feedback received from the editorial team, and for the opportunity to respond to the constructive points in our submitted manuscript. Please find below our point-by-point response to the feedback outlining, where relevant, the related changes we have made. We have uploaded revised versions of the manuscript as instructed, including both a clean copy and a track changes version, and additional supplementary files made accordingly.

Editor

Response

Thank you. We have confirmed that all style formatting and file naming adheres to the PLOS ONE templates as provided in the abovementioned pages. 

Response

We have included all minimal datasets used to produce the results presented in the manuscript as supporting information. 

3. In the online submission form, you indicated that Data cannot be shared publicly because of restrictions imposed by the ethics committee as most of these contain patient data, albeit de-identified, and it may be possible to determine the identity of participants. Should there be a request for data, this can be sent to the corresponding author (email: susanna.hutajulu@ugm.ac.id)

Response

We have now made necessary adjustment to the supplementary S1 File to also include minimal dataset to reproduce Table 1 and Table 2. All underlying data of the findings reported in the manuscript are now available in the supplementary information as required.

4. You have indicated that data is available from [email: mhrec_fmugm@ugm.ac.id]. Please can we ask you to provide us with a general contact email address for the data requests, so readers can request access in perpetuity. If a general email is not available, please provide a link to a website where readers can obtain access to data.

Response

Thank you for your comment. As stated in the original submission, all data access queries can be sent to the corresponding author (email: susanna.hutajulu@ugm.ac.id) or the institutional ethics committee (email: mhrec_fmugm@ugm.ac.id) at Universitas Gadjah Mada, Indonesia. 

5. Please respond by return e-mail with an updated version of your manuscript to amend either the abstract on the online submission form or the abstract in the manuscript so that they are identical. We can make any changes on your behalf.

Response

Thank you. We confirm that we have made no changes to our abstract and will respond with the updated version of our manuscript through a return e-mail along with this resubmission.

Response

Thank you for your advice. We have made sure that all references listed in the manuscript is correct and none of them have been retracted. We have added a reference as suggested by Reviewer #1 and address this in more details in the corresponding comment (Reviewer#1, comment number 3).

Reviewer #1

1. Please provide result of the indices of data quality of the PBCR.

Response

Thank you for your suggestion. We have now added S1 Table as supplementary to summarize the quality of the Yogyakarta PBCR data during the observed period.

2. Please provide a reference for the denominator.

Response

Thank you for your advice in elucidating the population reference used in calculating the age-standardized rate and the annual percent change of the CRC incidence in our study. Previously, we have stated in lines 187-9 under Materials and Methods that the number of populations of the Sleman, Kota Yogyakarta, and Bantul districts in 2014 was used as the denominator. Now we added the detailed number of populations in the S2 Table as supplementary.

3. Please mention the standard population with a reference used for the calculation of standardized rates.

Response

Thank you for your advice. We have now stated under Materials and Methods in line 191-2 that the world standard population as proposed by Segi, M (1960) was used in calculating the standardized rate. We have now also added this as reference number [31].

Additional remarks

We have made a correction in line 292 under Results, in which previously we refer to Fig 2 where it should be Fig 1 (middle). We confirm that this error does not affect other results presented in the manuscript.

Following our revisions, we hope the manuscript is suitable for publication in PLOS ONE.

Yours sincerely,

Susanna Hutajulu, MD, PhD

---

## [Editor Report · Decision Letter 1]

12 Mar 2024

Diverging likelihood of colon and rectal cancer in Yogyakarta, Indonesia: a cross sectional study

PONE-D-23-37906R1

Dear Dr. Hutajulu

We’re pleased to inform you that your manuscript has been judged scientifically suitable for publication and will be formally accepted for publication once it meets all outstanding technical requirements.

Kind regards,

Nontuthuzelo Iris Muriel Somdyala, Ph.D

Academic Editor

PLOS ONE